Vorinostat enhances chemosensitivity to arsenic trioxide in K562 cell line

Li Nainong nainli@aliyun.com
Guan Xiaoyan
Li Fang
Li Xiaofan
Chen Yuanzhong chenyz@mail.fjmu.edu.cn
Department of Hematology, Fujian Institute of Hematology, Fujian Provincial Key Laboratory on Hematology, Fujian Medical University Union Hospital , Fuzhou , PR China
Arany Praveen
Electronic publication date: 2015 May 28
Publication date: 2015
Volume: 3
Electronic Location ID: e962
Received 2015 Jan 16; Accepted 2015 Apr 26
Copyright: © 2015 Li et al.
Copyright year: 2015
Copyright holder: Li et al.
License: This is an open access article distributed under the terms of the Creative Commons Attribution License, which permits unrestricted use, distribution, reproduction and adaptation in any medium and for any purpose provided that it is properly attributed. For attribution, the original author(s), title, publication source (PeerJ) and either DOI or URL of the article must be cited.
License URL: https://creativecommons.org/licenses/by/4.0/

Keywords: Histone deacetylase inhibitors, Arsenic trioxide, K562 cell, Apoptosis

Funding: Fujian Provincial youth grant of Science and Technology 2008F3044 National and Fujian Provincial Key Clinical Specialty Discipline Construction Program This work is supported by a Fujian Provincial youth grant of Science and Technology (2008F3044) and sponsored by the National and Fujian Provincial Key Clinical Specialty Discipline Construction Program, P.R.C. The funders had no role in study design, data collection and analysis, decision to publish, or preparation of the manuscript.

==============================
Objective. This study aimed to investigate the chemosensitive augmentation effect and mechanism of HDAC inhibitor Vorinostat (SAHA) in combination with arsenic trioxide (ATO) on proliferation and apoptosis of K562 cells.

Methods. The CCK-8 assay was used to compare proliferation of the cells. Annexin-V and PI staining by flow cytometry and acridine orange/ethidium bromide stains were used to detect and quantify apoptosis. Western blot was used to detect expression of p21, Akt, pAkt, p210, Acetyl-Histone H3, and Acetyl-Histone H4 proteins.

Results. SAHA and ATO inhibited proliferation of K562 cells in an additive and time- and dose-dependent manner. SAHA in combination with ATO showed significant apoptosis of K562 cells in comparison to the single drugs alone (p < 0.01). Both SAHA and ATO alone and in combination showed lower levels of p210 expression. SAHA and SAHA and ATO combined treatment showed increased levels of Acetyl-Histone H3 and Acetyl-Histone H4 protein expression. SAHA alone showed increased expression of p21, while ATO alone and in combination with SAHA showed no significant change. SAHA and ATO combined therapy showed lower levels of Akt and pAkt protein expression than SAHA or ATO alone.

Conclusion. SAHA and ATO combined treatment inhibited proliferation, induced apoptosis, and showed a chemosensitive augmentation effect on K562 cells. The mechanism might be associated with increasing histone acetylation levels as well as regulating the Akt signaling pathway.

Introduction

Histone acetylation and deacetylation comprise one of the common modifications found in epigenetics. Histone acetylation helps transcription factors bind to DNA templates and activates transcription; however, histone deacetylation inhibits transcription (Ciro, Saverio & Pier, 2010; Wang, Allis & Chi, 2007; Reid & Gallais, 2009; Victoria, Jose & James, 2009). In 2006, the second generation of hydroxamic acids histone deacetylase inhibitor (HDACi) SAHA (suberoylanilide hydroxamic acid, Vorinostat) was approved by the FDA for clinical treatment of relapsed or refractory cutaneous T-cell lymphoma (Siegel et al., 2009). Chronic myeloid leukemia (CML) is a malignant clonal disease derived from hematopoietic stem cells, and it manifests itself through characteristic molecular markers: the bcr-abl fusion gene and the expression of p210 bcr-abl fusion protein. Tyrosine kinase inhibitors (TKI) are the first line of treatment for CML. Although TKIs target the bcr-abl gene, they are unable to eradicate the malignant clone completely. Moreover, drug resistance has been observed in clinical applications. To better eradicate the CML clone and overcome the drug resistance, we hypothesized that SAHA augments chemosensitivity when combined with Arsenic trioxide (ATO), another widely used chemoreagent for clinical treatment of hematologic malignancies. Thereby, we investigated the anti-leukemia effect of SAHA combined with ATO on the K562 cell line.

Material and Methods

Reagent and cell lines

SAHA was kindly provided by Dr. Defu Zeng (City of Hope National Medical Center, Duarte, California, USA). DMSO was diluted to a concentration of 10 mmol/L. ATO was purchased from Heilongjiang Harbin Medical University Pharmaceutical Co., Ltd., and diluted to a concentration of 2 mmol/L. Both SAHA and ATO were stored at −20 °C. RPMI 1640 was purchased from Gibco (Grand Island, New York, USA); fetal bovine serum was purchased from TBD (TBD Science, China); penicillin/streptomycin was purchased from Hyclone (Logan, Utah, USA); an apoptosis detection kit (Annexin-V-FITC, PI double staining) was purchased from Roche (Basel, Switzerland); Protein Extraction reagents were purchased from Wuhan Boster Biological Co., Ltd. (Wuhan, China); antibodies against c-abl, p21, Acetyl-Histone H3 and Acetyl-Histone H4, and other proteins were purchased from Cell Signaling Technology (Danvers, Massachusetts, USA). The K562 cell line was preserved and cultured in Fujian Institute of Hematology with a standard protocol as previously described (Liu et al., 2012; Lin et al., 2010).

Proliferation assay

A CCK-8 assay was used to detect cell line proliferation. A positive control, negative control and the drug treated group (SAHA: 0.25, 0.5, 1, 2 µmol/L; ATO: 1, 2, 4, 8 µmol/L) were detected. The assay was completed in triplicate for each group. Next, 5 × 104 cells/ml were seeded into a 96-well culture plate to a final volume of 100 µl. After 24, 48, and 72 h of treatment, 10 µl of CCK-8 was added into each well and incubated for 1–4 h at 37 °C in a 5% CO2 incubator. The absorbance at 450 nm was measured using a microplate reader. The inhibition rate was calculated using the following equation: Inhibition rate = [(control group − experimental group)/blank group] ×100%. A proliferation curve was plotted based on the drug concentration and the proliferation inhibition rate. The Q value was calculated to measure the combined effect of SAHA and ATO. Q was calculated using the following equation: Q = E(ab)/(Ea + Eb − Ea × Eb) as previously described (Liu et al., 2012; Lin et al., 2010). Generally, Ea and Eb are the single drug inhibition rates; E(ab) is associated with the inhibition rate of SAHA and ATO. Q > 1.15 was considered to be a synergistic effect, 0.85 > Q > 1.15 an additive effect, and Q < 0.85 an antagonistic effect.

Annexin-V and PI staining

Apoptosis was detected by Annexin-V and PI staining. Generally, K562 cells (1 × 106) and the appropriate drugs were co-incubated for 48 h. Cells were harvested after a single wash with PBS. Next, binding buffer (100 µl) was added to resuspending cells. Annexin-V (2 µl) and PI (2 µl) were added, and the cells were incubated at room temperature for 10–15 min in the dark. K562 cell were detected by flow cytometry. Annexin-V−/PI− cells were living cells, and Annexin-V+/PI− cells were early apoptotic cells. Annexin-V+/PI+ were late apoptotic cells. This experiment was repeated three times.

AO/EB fluorescence staining

K562 cells and the drugs were co-incubated as described above. Suspended cells (95 µl, 5 × 106/ml) were mixed with AO/EB (5 µl). Immediately after mixing, one drop of suspended cells was placed on a clean glass slide and observed by fluorescence microscopy at an excitation wavelength of 490 nm. Cells with green fluorescence in the nucleus and cytoplasm were normal cells; cells with yellow–green fluorescence in the nucleus or cytoplasm were apoptotic cells; cells with red fluorescence in nucleus were necrotic cells.

Western blot

K562 cells were co-cultured as described above. After the cells were lysed, the supernatant was collected and quantified. Total protein (40 µg) was transferred to a PVDF membrane after SDS-PAGE electrophoresis. After blocking at room temperature for 2 h, the primary antibodies (against p210, p21, Acetyl-Histone H3, Acetyl-Histone H4, Akt and pAkt) were added and incubated at 4 °C overnight. After washing, the secondary antibody (horseradish peroxidase-conjugated anti-mouse IgG) was added and incubated for 2 h. The chemiluminescence reaction was performed. β-actin was used as an internal control.

Statistical analysis

Statistical analysis was performed with SPSS 11.5. Means were compared by the Dunnett-t test, with P < 0.05 considered to be a significant difference.

Results

SAHA in combination with ATO showed an additive effect on the inhibition of proliferation

Compared with the control group, SAHA and ATO inhibited proliferation of K562 cells in a time- and dose-dependent manner. After the cells were treated with SAHA and ATO, cell proliferation was significantly inhibited (Fig. 1). SAHA in combination with ATO showed an additive effect on the inhibition of proliferation (Table 1).

Figure 1 SAHA and ATO alone showed inhibition of proliferation on K562 cell line.

5 × 104 cells/ml were seeded to a 96-well culture plate in a final volume of 100 µl. After 24, 48, and 72 h of treatment, 10 µl of CCK-8 was added into each well and incubated for 1–4 h. The absorbance at 450 nm was measured using a microplate reader. Inhibition rate = [(control group − experimental group)/blank group] ×100%. (A) The proliferation curve of SAHA alone is shown. (B) The proliferation curve of ATO alone is shown.

Table 1 SAHA in combination with ATO showed an additive effect on the inhibition of proliferation.

5 × 104 cells/ml were seeded to a 96-well culture plate in a final volume of 100 µl. Different concentration of drugs were applied. Inhibition rate was calculated as Fig. 1. Q was calculated using the following equation: Q = E(ab)/(Ea + Eb − Ea × Eb). Ea and Eb are the single drug inhibition rates; E(ab) is associated with the inhibition rate of SAHA and ATO. Q > 1.15 was considered to be a synergistic effect, 0.85 > Q > 1.15 an additive effect, and Q < 0.85 an antagonistic effect.

Concentration (µmol/L)	Inhibition (%)	Q value	Additive effect	
SAHA	ATO				
0	0	0			
0.25	0	45.33			
0.5	0	62.24			
1	0	76.01			
2	0	95.84			
0	1	3.7			
0	2	14.07			
0	4	76.67			
0	8	96.99			
0.25	1	49.91	0.906	+	
0.25	2	55.75	1.051	+	
0.25	4	84.08	0.964	+	
0.5	1	63.85	1.003	+	
0.5	2	70.88	1.049	+	
0.5	4	85.41	0.937	+	
1	1	84.75	1.102	+	
1	2	87.19	1.098	+	
1	4	96.01	1.017	+	

SAHA in combination with ATO showed more apoptosis than the single drugs alone

Annexin-V-FITC/PI double staining flow cytometry can distinguish early apoptotic cells, late apoptotic cells and necrotic cells. In the lower left quadrant are normal cells; in the lower right quadrant are early apoptotic cells; in the upper right quadrant are late apoptotic cells; in the upper-left quadrant are necrotic cells. The results showed that 95% of the cells in the control group were living cells; the single drug apoptosis rates increased with increasing drug concentration. The K562 cells apoptosis rates were 10.85 ± 0.65%, 29.65 ± 1.75%, and 84.00 ± 0.80%, 48 h after treatment with SAHA (2 µmol/L), ATO (8 µmol/L) and 8 µmol/L ATO combined with 2 µmol/L SAHA, respectively. SAHA in combination with ATO showed significant apoptosis of K562 cells compared to the single drugs alone (p < 0.01, Fig. 2A).

Figure 2 Apoptosis rate after treatment.

(A) Apoptosis was detected by Annexin-V and PI staining. K562 cells (1 × 106) and the drugs were co-incubated for 48 h. Cells were harvested after a single PBS wash. 100 µl of binding buffer was added to resuspend the cells. Annexin-V (2 µl) and PI (2 µl) was added. After an incubation at room temperature for 10–15 min in the dark, K562 cells were detected using a Flow Machine. The apoptosis rates after different treatments are shown (SA2, 2 µmol/L SAHA; SA4, 4µmol/L SAHA; AS4, 4 µmol/L ATO; AS8, 8 µmol/L ATO; SA2 + AS8, 2 µmol/L SAHA in combination with 8 µmol/L ATO). (B–E) Apoptosis observations using fluorescence microscopy 48 h after treatment; (×400 magnification). K562 cells and the drugs were co-incubated as described in Fig. 2. Suspended cells (95 µl, 5 × 106/ml) were mixed with AO/EB (5 µl). Immediately after mixing, one drop of suspended cells was placed on a clean glass slide and observed by fluorescence microscopy at an excitation wavelength of 490 nm. Cells with green fluorescence in the nucleus and cytoplasm were normal cells; cells with yellow–green fluorescence in the nucleus or cytoplasm were apoptotic cells; cells with red fluorescence in the nucleus were necrotic cells. (B) control; (C) 2 µmol/L SAHA; (D) 8 µmol/L ATO; (E) 2 µmol/L SAHA in combination with 8 µmol/L ATO.

The apoptosis results were confirmed by AO/EB staining. In the control group, K562 cells showed uniform cell size and morphology and homogeneous green fluorescence in the nucleus and cytoplasm. In contrast, 48 h after SAHA and ATO combined treatment, we found differences in cell size and morphology. The nucleus was dense, showing yellow–green fluorescent debris. Additionally, SAHA in combination with ATO showed more apoptosis characteristics than the single drugs alone (Figs. 2B–2E).

SAHA in combination with ATO showed pronounced changes in protein expression

Both SAHA and ATO alone and in the combination group showed lower levels of p210 expression and. The SAHA group and the combination group showed increased levels of Acetyl-Histone H3 and Acetyl-Histone H4 protein expression. SAHA alone showed increased levels of p21 WAF1/CIP1 expression, while ATO alone and in the combination group showed no significant changes. The combination group showed lower levels of Akt and pAkt protein expression than SAHA or ATO alone (Fig. 3).

Figure 3 SAHA in combination with ATO showed pronounced changes in protein expression.

K562 cells were co-cultured as described above. p210, p21, Acetyl-Histone H3, Acetyl-Histone H4, Akt and pAkt were detected by Western blot. β-actin was used as an internal control. (C, control S, 2 µmol/L SAHA; A, 8 µmol/L ATO; S+A, 2 µmol/L SAHA in combination with 8 µmol/L ATO.)

Discussion

There are a variety of HDACIs undergoing clinical trials for the treatment of solid tumors and hematological malignancies, especially for the treatment of cutaneous T-cell lymphoma (CTCL), peripheral T-cell lymphoma, and Hodgkin’s lymphoma, which have shown promising results (Duvic et al., 2007). As a novel HDACI, SAHA was approved by the FDA for the treatment of progressive clinical recurrence of CTCL.

ATO is an important medicine for treating leukemia (particularly for acute promyelocytic leukemia, APL). Additionally, chemotherapy regimens based on ATO are a popular research topic. In recent years, the mechanism and clinical applications of ATO have been reported, with induction of oxidative stress and DNA damage included (Sanjay, Yedjou & Tchounwou, 2014; Miller et al., 2002). Our study mainly focused on the Akt signal pathway. Huang reported that ATRA and arsenic compound-based combination therapy was effective in re-inducing morphological remission in relapsed patients with APL who had previous exposure to ATRA and arsenic compounds, thus producing low molecular remission rates and high risk of secondary relapse (Lu et al., 2014). Interestingly, we also found chemosensitive augmentation of SAHA in the NB4 cell line (data not shown), which indicates that combined SAHA and ATO therapy could have clinical applications in the treatment of leukemia.

The CCK8 results show that individual treatments with SAHA or ATO inhibited K562 cell proliferation in a time- and dose-dependent manner. The combination treatment showed an additive effect in inhibiting proliferation. Additionally, experimental results showed that at certain combined concentrations of SAHA and ATO, there was a greater level of apoptosis than the single drugs alone, which induced apoptosis over 80%. These results indicated that the combined treatment might have beneficial effects including improved chemotherapeutic effects and minimized side effects.

In this study, both SAHA and the combination group induced increased expression of Acetyl-Histone H3 and Acetyl-Histone H4 when compared to ATO and the control. This outcome indicates that SAHA can significantly increase histone acetylation levels and activate transcription, thereby inducing hematological tumor cell apoptosis (Reid & Gallais, 2009; Victoria, Jose & James, 2009).

p21 is an important cell cycle regulatory protein involved in cell growth, differentiation, aging and death. p21 plays a negative role in the regulation of the cell cycle by acting on CDKI. p21 expression was upregulated after treatment with SAHA compared to ATO and the control, thereby blocking the cell cycle and inducing apoptosis. Interestingly, combination therapy did not induce expression of p21. This mechanism needs to be studied further.

Akt is a serine/threonine protein kinase, which plays a major role in the anti-apoptotic pathway in apoptosis signaling (Osaki, Oshimura & Ito, 2004). We found that both the SAHA group and ATO group downregulate Akt and pAkt expression. Compared to the single drug treatment, the combination group downregulated Akt and pAkt expression drastically, indicating that the mechanism may be associated with the regulation of the Akt signaling pathway.

p210 fusion protein is the primary cause of CML pathogenesis and progression. Furthermore, the p210 fusion protein is the source of the abnormal phenotype of chronic phase CML. We found that both the SAHA group and the ATO group downregulate p210 fusion protein expression. Also, the combination group showed downregulation of p210 protein. Given the dosage which we applied by proliferation experiments, both SAHA and ATO in the concentration are very effective in downregulating the p210 fusion protein expression. It could be more effective by the combination. Taken together, these results indicated that both SAHA and ATO can induce cell apoptosis and inhibition of cell proliferation by downregulating the characteristic p210 fusion protein in K562 cells. Combining SAHA with ATO also lowered the expression of p210, which resulted in increased apoptosis.

In summary, SAHA and ATO can significantly inhibit the proliferation of K562 cells and induce apoptosis. Combined treatment can enhance apoptosis effects within a certain concentration range. The mechanism may be associated with increased histone acetylation and the regulation of the Akt signaling pathway. This study may provide a new combination chemotherapy regimen for the treatment of CML. Further in vivo experiments and animal model should be set up to test the enhanced chemosensitivity effect by combining SAHA and ATO. We also need to observe the side effects by combining SAHA and ATO and prepare data for translational medicine work.

Additional Information and Declarations

Competing Interests

Author Contributions

The authors declare there are no competing interests.

Nainong Li conceived and designed the experiments, analyzed the data, wrote the paper, reviewed drafts of the paper.

Xiaoyan Guan performed the experiments, analyzed the data, prepared figures and/or tables.

Fang Li for the resubmission work, revision exp.

Xiaofan Li wrote the paper.

Yuanzhong Chen contributed reagents/materials/analysis tools, reviewed drafts of the paper.

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
