# Peer review of "Vorinostat enhances chemosensitivity to arsenic trioxide in K562 cell line"

_PeerJ, doi:10.7717/peerj.962_

## Round 0.1 · original submission · Major Revisions

While this is an interesting study suitable for this journal, kindly address the relevant issues raised by the reviewers for further consideration..

·

Basic reporting

No comments

Experimental design

1. In figure 3, the western blot shows reduction in the Akt protein levels. However that is insufficient to show changes in activity. It would be better to show the decrease in activity by another method, such as reduced phosphorylation of targets or a reporter assay.

2. All assays shown here are on a cell-line. The effect of the two drugs shown here is mostly additive. The two drugs together can be toxic in a physiological context, which may be responsible for increased cell death. Therefore in order to say that the effect is better in combination, in vivo experiments are essential.

Validity of the findings

The manuscript reports the observations of the study fairly well. However, there is no physiological relevance of these findings. The two drugs SAHA and ATO show an additive effect on cells which could be the effect of increased toxicity.
Both the drugs used ATO and SAHA can cause global epigenetic changes in methylation and histone acetylation respectively. If given in combination to a mouse model, what would be the effect? Will the animal sustain the drug levels and what would be the side effects? These points need consideration.

Additional comments

1. Figure 3 shows that p210 levels are lost with any one drug treatment also. Lines 217-224 claim that the combination of drugs work better than individual drugs which is not reflected in the figure. This discrepancy needs to be fixed.
If both the drugs individually can down regulate p210 levels, what is the significance of the combination?

·

Basic reporting

The manuscript by Li et al addresses the synergistic effect of HDAC inhibitor vorinostat (SAHA) and arsenic trioxide (ATO) on proliferation and apoptosis of K562 cells. Flow cytometric analysis of annexin V/PI and AO/EB staining showed that apoptotic cell death was significantly enhanced after this combined treatment. SAHA alone showed increased expression of p21, while ATO alone and in combination with SAHA showed no significant change. However, SAHA and ATO combined therapy showed lower level of Akt expression. The authors suggest that the SAHA and APO combined treatment induces apoptosis by increasing histone acetylation levels as well as activation of Akt signaling pathway. Numerous HDAC inhibitors, either alone or in combination with other chemotherapeutic agents are in clinical development. The combination of HDAC inhibitor with DNA damaging agent (ATO) may provide an improved therapeutic approach. However, there already exists literature addressing the effects of combined treatment of SAHA with ATO on tumor growth. How ATO enhance the apoptosis caused by SAHA, is not clear from this paper. The current manuscript makes no solid attempt to provide a clear mechanism to account for combined apoptotic effects.

Experimental design

The difficulty with the paper as presented is that it is not clearly written nor the figures are presented as they should have been. It limits the interest of readers to a great extent. Arsenic trioxide is known to induce oxidative stress and DNA damage. There has not been a single attempt to address the issue of DNA damage in this paper and not even a single article cited in this regard.

Validity of the findings

Major Comments
1. Activation of Akt pathway could be suggested using levels of phosphoAkt and the key players of the pathway.
2. The level of p53 has not been shown in Figure 3 but has been mentioned by authors in the abstract.
3. As authors suggest that the mechanism of apoptosis is through acetylation of histone but increased levels of acetyl histone H3 and H4 is not clear from the western blot (Fig3)
Minor comments
4. A figure legend of the Table 1 is missing.
5. Graphical representation of the Figure 1 could have been A) SAHA alone B) ATO alone C) the combination of ATO and SAHA.
6. It would have been more convincing to see the Flow cytometric data presented in Fig. 2A with representative flow histograms than the bar diagrams alone.

Reviewer 3 ·

Basic reporting

No Comments

Experimental design

No Comments

Validity of the findings

The study by Li et al investigated the anti-leukemia effect of combining Vorinostat (SAHA) and arsenic trioxide (ATO) on K562 cells. The results showed that the combined treatment (SAHA plus ATO) may exert the anticancer activity through inhibiting cell proliferation, inducing apoptosis, and altering the expression of proteins critical for chronic myeloid leukemia (CML) proliferation. The anti-CML effects are promising and worth further investigation. However, the report is descriptive and lack of mechanistic studies.

The major concerns are:
1. Fig. 1 shows that treatment of K562 cells with 2 uM SAHA (Fig. 1A) or 8 uM ATO (Fig. 1B) for 48 h almost completely inhibited cell proliferation as manifested by the proliferation inhibition rate nearing 1.0. However, Fig. 2 reveals that the same treatment respectively induced 10% and 30% cell apoptosis. What is the possible explanation for this discrepancy? Do SAHA and ATO, when single-treated, inhibit K562 cell proliferation by means other than inducing apoptosis? Complete inhibition of cell proliferation or induction of 100% apoptosis, which one is a preferred strategy in terms of CML chemotherapy?

2. In “Abstract” and “Discussion”, the authors consider the mechanism associated with the enhanced K562 sensitivity to combined treatment of SAHA and ATO to be “increased histone acetylation and the activation of the Akt signaling pathway”. However, Fig. 3 reveals that the levels of acetyl-Histone H3 and Akt are prominently decreased in the combined treatment (S+A) comparing to SAHA alone (S).

3. In the analysis of protein expression shown in Fig. 3, all three treatments (2 uM SAHA, 8 uM ATO, and 2 uM SAHA plus 8 uM ATO for 48 h) completely inhibited cell proliferation. Moreover, the combined treatment also induced nearly 90% cell apoptosis. Would it be possible the expression levels of proteins critical for cell proliferation and survival in combined-treated cells merely reflect the apoptotic status of the treated cells not a primary action mechanism of drugs?

---

## Round 0.2 · Minor Revisions

Please address the concern raised by reviewer 1 and kindly mention the limitation if this study in the discussion as suggested.

·

Basic reporting

No comments

Experimental design

1. The authors have shown change in activity of Akt by including a pAkt western blot in fig 3. This is a revision which was suggested previously and has been addressed.

Validity of the findings

No comments

·

Basic reporting

No comments

Experimental design

No comments

Validity of the findings

No Comments

Additional comments

Color is not needed to convey the scientific message of Figure 1 so a black and white version is better.

---

## Round 0.3 · accepted · Accept

Thank you for appropriately addressed the concerns raised by the reviewers and Congratulations! We look forward to working with you in future submissions.